# Comparative Tensile Properties and Collagen Patterns in Domestic Cat (*Felis catus*) and Dog (*Canis lupus familiaris*) Ovarian Cortical Tissues

**DOI:** 10.3390/bioengineering10111285

**Published:** 2023-11-04

**Authors:** Jennifer B. Nagashima, Shoshana Zenilman, April Raab, Helim Aranda-Espinoza, Nucharin Songsasen

**Affiliations:** 1Center for Species Survival, Smithsonian National Zoo and Conservation Biology Institute, 1500 Remount Rd., Front Royal, VA 22630, USA; songsasenn@si.edu; 2College of Veterinary Medicine, Cornell University, 144 East Ave, Ithaca, NY 14850, USA; 3College of Veterinary Medicine, Michigan State University, 784 Wilson Rd., East Lansing, MI 48824, USA; 4Fischell Department of Bioengineering, University of Maryland, 3108 A. James Clark Hall, College Park, MD 20742, USA; helim@umd.edu

**Keywords:** ovary, domestic cat, domestic dog, rigidity, collagen, matrix metalloproteinase, in vitro culture

## Abstract

The importance of the ovarian extracellular environment and tissue rigidity on follicle survival and development has gained attention in recent years. Our laboratory has anecdotally observed differences in the rigidity of domestic cat and dog ovarian cortical tissues, which have been postulated to underlie the differences in in vitro culture responses between the species, wherein cat ovarian tissues display higher survival in extended incubation. Here, the tensile strengths of cat and dog ovarian cortical tissues were compared via micropipette aspiration. The underlying collagen patterns, including fiber length, thickness, alignment, curvature, branch points and end points, and overall tissue lacunary and high-density matrix (HDM) were quantified via picrosirius red staining and TWOMBLI analysis. Finally, we explored the potential of MMP (−1 and −9) and TIMP1 supplementation in modulating tissue rigidity, collagen structure, and follicle activation in vitro. No differences in stiffness were observed between cat or dog cortical tissues, or pre- versus post-pubertal status. Cat ovarian collagen was characterized by an increased number of branch points, thinner fibers, and lower HDM compared with dog ovarian collagen, and cat tissues exposed to MMP9 in vitro displayed a reduced Young’s modulus. Yet, MMP exposure had a minor impact on follicle development in vitro in either species. This study contributes to our growing understanding of the interactions among the physical properties of the ovarian microenvironment, collagen patterns, and follicle development in vitro.

## 1. Introduction

In the last decade, there has been growing understanding of the importance of the physical microenvironment on the survival, growth, and developmental competence of ovarian follicles. Such understanding is key to developing technologies to grow immature follicles in vitro to produce fertilizable oocytes. Although live offspring have been produced from oocytes from in vitro-grown laboratory mouse follicles [1,2,3], many of the in vitro culture techniques have had more limited success when applied to larger mammalian systems [4,5,6,7,8,9,10]. However, this technology would be invaluable to both human fertility preservation and endangered species genetic rescue efforts.

Domestic cats and dogs are excellent large mammalian models for both human biomedical research [11,12] and for wild felid and canid reproductive studies [13], respectively. Previously, our laboratory has compared various in vitro culture systems on both ovarian-tissue-enclosed and isolated preantral follicle growth. We have observed that, in both conventional agarose gel culture and microfluidic incubation, and under different media conditions [14], the survival and development of domestic dog follicles in cultured tissue is much poorer compared with cat follicles. Anecdotally, this has been attributed to the much more rigid cortical tissue in dogs. This is supported by the findings of our laboratory [15] and others [16,17] that isolation of preantral dog follicles from cortical tissue does allow for follicle survival and even antral cavity formation in vitro. Young’s modulus values and metrics of mechanical elasticity or stiffness of a material or tissue have been measured through a variety of techniques to identify range values for ovarian tissues in women (2–15 kPa [18,19]), mice (0.5–10 kPa [20]), and cattle (0.7–11.6 kPa [21]). More recently, the elastic modulus of domestic cat ovarian tissues was evaluated via atomic force microscopy (AFM) and found to have an apparent elastic modulus (“apparent” because AFM probe indentation rates can affect the value) range of 0.2–17.4 kPa, with increased cortical rigidity in older animals (>6 mo) compared with younger individuals [22]. There is as yet no published information on the physical rigidity of domestic dog ovarian tissue.

It has been posited that a less rigid microenvironment is more permissive to ovarian follicle growth, especially during the transition from the primordial to the primary stage [23]. Indeed, Woodruff et al. [24] demonstrated that intra-ovarian murine follicles activate in a wavelike pattern from the rigid cortical region to the more permissive medulla of the ovary. To elucidate the interaction between tissue stiffness and folliculogenesis, there has been increasing focus on the ovarian extracellular matrix, or ECM. Ovarian cortical tissue ECM contains multiple collagen types (I, III, IV, and VI), as well as glycoproteins, proteoglycans, and various ECM-associated proteins [25,26,27]. On the whole, increased collagen density is positively correlated with increased tissue rigidity. This has been demonstrated in murine ovaries, where higher collagen matrix density in older animals’ tissues was associated with increased tissue stiffness [28]. Conversely, while focusing on type I collagen specifically, Stewart et al. found no significant correlation between collagen content and ovarian tissue stiffness in the domestic cat, though hyaluronic acid, a glycosaminoglycan, was positively associated [22]. However, differences in collagen patterns can also have a substantial impact on rigidity. In bovine bone, fiber orientation and alignment have been shown to contribute significantly to tissue tensile strength [29]. In decellularized bovine ovaries, cortical tissues are characterized by radially aligned collagen fibers within pore (areas absent of ECM) walls, whereas medullary regions are more porous with anisotropically-arranged collagen [30]. Cumulatively, it is clear that understanding the contribution of ECM architecture to tissue tensile properties is important in elucidating the mechanics of ovarian folliculogenesis and our ability to recapitulate it in vitro for large mammalian species.

Microenvironment permissivity can also be modulated by the follicles themselves, wherein activated follicles are thought to remodel their encompassing tissue via local enzyme secretion, including matrix metalloproteinases [31,32]. Matrix metalloproteinases, or MMPs, are a family of over 20 zinc-containing proteases that are best known for their ability to degrade extracellular matrix proteins, including collagen, elastin, and gelatin [33]. Both MMPs and tissue inhibitors of MMPs (TIMPs) play active roles in the ovary during folliculogenesis, including being involved in the tissue rupture process of ovulation [34]. MMP protein expression during cat folliculogenesis has previously been described, wherein MMP1 and MMP9 reach peak protein expression in early antral-stage follicles, primarily in the granulosa cells, and immunoreactivity for TIMP1 is maximal in secondary-stage follicles [35]. While less is known about the endogenous expression of MMPs and TIMPS in dog tissues, they represent an interesting avenue for altering the anecdotally tough cortical tissue.

In this study, we aimed to (1) compare the normal range of ovarian tissue tensile strength in cats and dogs via the micropipette aspiration technique [36,37] to improve our understanding about the differences in native follicular environments between the species and (2) characterize patterns in ovarian cortical tissue collagen fibers in both species to contribute to the growing understanding of how various ECM properties contribute to tissue viscoelasticity. Finally, we explored the potential of MMP (−1 and −9) and TIMP1 supplementation in modulating ovarian tissue rigidity, collagen structure, and primordial follicle activation in vitro.

## 2. Materials and Methods

Unless otherwise stated, all chemicals were purchased from Sigma Aldrich (St. Louis, MO, USA) and base media from Irvine Scientific (Irvine, CA, USA).

### 2.1. Ovarian Tissue Isolation

Domestic cat and dog ovaries were collected opportunistically at local spay clinics from routine ovariohysterectomies and transported to the laboratory in L-15 medium containing 10 mM HEPES, 100 µg/mL penicillin G sodium, and 100 µg/mL streptomycin sulfate on ice. Details on individuals, including ages and dog breeds (where known) are provided in Appendix A. Tissues were processed within 10 h of surgery. Cortical slices (~1 mm thick) were dissected from each ovary’s surface, then trimmed into 2 mm^2^ sections for micropipette aspiration evaluation and cortical tissue culture. Fresh tissues from each individual were also fixed to evaluate follicle populations and morphology.

### 2.2. Micropipette Aspiration Tensile Testing

Ovarian cortical (fresh and cultured) tissues were gently sandwiched between two glass slides using a line of putty (STA PUT Plumber’s putty, Hercules) to prevent tissue compression and allow for immersion in Leibovitz’s medium (L-15) during analyses. Tissues were brought to room temperature and tensile evaluations were completed within an hour of dissection.

A 0.5 mm inner-diameter, 1 mm outer-diameter glass microcapillary was pre-equilibrated in PBS at room temperature for 15 min, then loaded onto a micropipette holder attached to a pneumatic transducer tester (Fluke biomedical, DPM1B). A small volume of L-15 (~10 µL) was loaded into the microcapillary. The sandwiched tissue was then visualized under an inverted microscope (Olympus IX81) equipped with a camera (Rolera-MGi 1394, QImaging, (Teledyne Photometrics) Tucson, AZ, USA) and the microcapillary was threaded through the slides and positioned near the edge of the tissue (set up in Appendix A). An image was taken of the capillary (time 0, pressure 0); then, the stage was moved to bring the tissue into direct contact with the glass capillary (Figure 1a). Negative pressure (<5 mmHg) was then applied to the tissue using the pneumatic transducer tester to create a seal between the tissue and the glass capillary. Once established, sets of time-lapse series of images were taken (each series 2 min, 40 images per pressure or 3 s per image) as the negative pressure was increased stepwise in increments of 25 mmHg (e.g., 5 to 30 to 55 to 80 mmHg) to assess the amount of deformation of the tissue under increasing negative pressure (Figure 1a). Two to three areas (one from each of the two tissue pieces per animal) were assessed per treatment and individual for cats and dogs, respectively. Measurements were taken up to at least 80 mmHg, unless the tissue broke under the pressure before that point, which occurred in 4 cat sample replicates. For these evaluations, Young’s modulus values were determined based on the measurements taken prior to the break.

Young’s moduli for each set of tissues were calculated as previously described [38]. Briefly, the maximum distension of the tissue within the lumen of the microcapillary of known radius was plotted against the corresponding pressure, and the slope of this relationship was utilized to determine the Young’s modulus kPa, using the infinite homogeneous half-space model [39].
(1)$$\;\;E=\frac{3∆P}{2\pi \delta }\phi \left(\eta \right)$$
(2)$$\delta =\frac{d}{R_{i}}$$
(3)ϕη=121+η1+η2ln⁡(8η)
(4)$$\eta =(\frac{R_{e}-R_{i}}{R_{i}})$$
where *E* is the Young’s modulus (Equation (1)), Δ*P* is the change in aspiration pressure, *R_e_* is the micropipette outer radius, *R_i_* is the micropipette inner radius, and *d* is the distance or length of tissue distension into the pipette. Equation (2) is the wall function, defined by the boundary conditions at the contact zone between the tissue and the micropipette, Equation (3), and the external micropipette radius. Theret et al. used two sets of boundary conditions: force and punch models. The differences are very small and the force model was applied in the analyses of this study.

### 2.3. Histological Assessments

Ovarian cortical tissues were fixed in 4% paraformaldehyde overnight at 4 °C, then stored in 70% ethanol prior to being embedded in paraffin. Paraffin blocks were cut in 6 µm thick sections for the evaluation of collagen patterns via picrosirius red staining, as previously described [40]. Briefly, slides were deparaffinized and serially dehydrated, then exposed to a 0.1% solution of Direct Red 80 (Sigma Aldrich, St. Louis, MO, USA) in saturated picric acid for 40 min at room temperature, then exposed to 4 washes of 5% glacial acetic acid in water for 7 min. Following ethanol washes and Slide Brite (Biocare Medical, Pacheco, CA, USA) exposure, slides were mounted with a glass coverslip. Sections containing the largest cross-section of tissue per treatment/individual were imaged at 40x magnification on an EVOS2 Fl Auto2. Subsequent analysis of extracellular matrix patterns, including total fiber length, end points (count of the number of fiber ends), branch points (number of intersections of fibers identified in the image), alignment, hyphal growth unit (measurement of end points/fiber length), curvature (mean change in angle moving along individual fibers), lacunarity (measurement of gap number and size in the matrix), high-density matrix (HDM), and estimated fiber thickness were performed using ImageJ plugin TWOMBLI (The Workflow Of Matrix BioLogy Informatics v.1, gitub.com/wershofe/TWOMBLI, accessed on 4 May 2022) [41] in ImageJ. The TWOMBLI parameters were as follows: contrast saturation = 0.35, min. line width = 10, max. line width = 15, min. curvature window = 30, max. curvature window = 30, min. branch length = 10, max. display high-density matrix (HDM) = 175. Average fiber length was determined by dividing total fiber length by 0.5 (branch points + end points), and estimated thickness by dividing the HDM matrix by total fiber length [41].

Tissues were also evaluated via hematoxylin and eosin staining for ovarian follicle density. Morphologically normal follicle counts were evaluated in every 10 sections through 360 µm tissue depth. Primordial-stage follicles were defined as a centralized oocyte surrounded by a single layer of flattened granulosa cells. Primary follicles contained a single layer of cuboidal granulosa, and secondary-stage follicles had at least two layers of granulosa cells and, together with primary-stage follicles, were considered “preantral”. Follicles observed to have a combination of flattened and cuboidal granulosa cells in a single layer were classified as activated, “transitional” follicles. Only follicles with visible nuclei were counted. Follicles with pyknotic nuclei were considered non-normal morphology or “atretic”. Comparisons of normal follicle density among treatment groups were performed by comparing the number of morphologically normal follicles of each stage (primordial, transitional, primary, and secondary) out of the total tissue area (mm^2^). For each individual and treatment group, follicle diameters were measured in at least 4 sections with the highest follicle counts.

### 2.4. MMP Exposure Trial

To explore the effect of MMP supplementation on dog and cat ovarian tissue tensile strength and collagen patterns, we first identified appropriate concentrations for MMP1, MMP9, and TIMP1 supplementation. The selection of MMPs in the current study was based on previous work with cats, which demonstrated gene expression of both MMP1 and MMP9 and of TIMP1 in all follicle stages [42], with MMP1 and MMP9 protein expression in both granulosa and oocytes in all follicle stages, as well as in stromal cells [35]. Similar information on the specific MMPs produced by the various ovarian compartments is not yet known for dogs. Previous research with women has also established that protein concentrations of MMP9 [43,44] and TIMP1 [44,45] are 100× and 6× higher in circulation compared with follicular fluid, respectively. For MMP1 in humans, reported serum values in rheumatoid arthritis patients [46,47] are similar to follicular fluid concentrations (4.6–6.5 ng/mL [48]). Because our experiment was intended to evaluate potential toxicity in long-term exposure, only domestic cat ovaries were evaluated because they have been demonstrated to survive longer in in vitro culture compared with domestic dog ovaries [14]. Cat ovarian cortical tissues were cultured for 14 days on an agarose gel block in either 0, 5, 10, 50, 100, or 500 ng/mL rhMMP1 (ab124850, *n* = 3 cats), rhTIMP1 (ab82104, *n* = 3 cats), or rhMMP9 (ab155704, *n* = 2 cats), with MMPs being prepared per the manufacturer’s instructions. Briefly, MMP1 and MMP9 were diluted in TCMB (50 mM Tris, 10 mM CaCl_2_, 150 mM NaCl, 0.05% Brij-35 (*w*/*v*), pH 7.5) buffer, then activated prior to supplementation via incubation with 1mM p-aminophenylmercuric acetate (APMA) for >1 or 8 h, respectively, at 37 °C. At the end of culture, tissues were fixed and morphology and follicle populations were evaluated via hematoxylin and eosin staining. The highest concentrations yielding surviving preantral follicles following the 14 d culture were selected for this study (Appendix A).

Based on the initial MMP and TIMP testing, six different supplemental treatment groups were evaluated and compared with fresh tissue: control (no supplementation), 30 ng/mL MMP1, 10 ng/mL TIMP1, 10 ng/mL MMP9, 30 ng/mL MMP1 + 10 ng/mL TIMP1 (1p1), and 30 ng/mL MMP1 + 10 ng/mL TIMP1 + 10 ng/mL MMP9 (all). Four pieces of ovarian tissue were cultured in each treatment group per animal, plus four pieces fixed upon collection as fresh controls. Domestic dog (*n* = 7) and cat (*n* = 6) ovarian cortical tissues were cultured for 5 or 14 days, with MMP and TIMP supplementations during only the first 5 days of culture (with fresh supplementation in full medium exchange feedings on days 2 and 4 of culture). After 5 days, the culture medium was exchanged for a fresh control medium for all treatment groups, and half of the culture medium was subsequently exchanged for fresh every other day until day 14. At the termination of culture, two pieces each were utilized for evaluation of rigidity (via micropipette aspiration), follicle population, and collagen density (histological assessments). A separate culture of 8 cat and 7 dog tissues was incubated for 5 days in a subset of the treatment groups for gene expression analyses.

### 2.5. Ovarian Tissue Culture

Domestic cat tissue was cultured in modified Eagle’s medium (MEM) supplemented with 4.2 µg/mL insulin, 3.8 µg/mL transferrin and 5 ng/mL selenium, 2 mM L-glutamine, 100 µg/mL penicillin G sodium and streptomycin sulfate, 0.05 mM ascorbic acid, and 1 mg/mL BSA. Further, 10 ng/mL FSH and 100 ng/mL EGF, which have previously been utilized for cat ovarian tissue culture [49], were supplemented. Dog ovarian cortical tissue was cultured in alpha-MEM supplemented with 1 mg/mL BSA, 4.2 µg/mL insulin, 3.8 µg/mL transferrin and 5 ng/mL selenium, 2 mM glutamine, 10 IU/mL of penicillin G and 10 µg/mL of streptomycin, and 10 µg/mL follicle-stimulating hormone (FSH) [50]. For both species, tissue sections for each animal were cultured on a 1.5% agarose gel block [49] at 38 °C and 5% CO_2_, with half of the medium being exchanged for fresh every other day.

### 2.6. Gene Expression

Expression of various genes associated with oocyte health (growth differentiation factor 9 [*GDF9*] [51]), development (zona pellucida proteins [*ZPA, ZPB*, and *ZPC*], vascular endothelial growth factor [*VEGF*] [52]), and survival/proliferation (proliferating cell nuclear antigen [*PCNA*], caspase 3 [*CASP3*]) relative to the reference gene beta actin [*ACTB*] [53] (Table 1) were evaluated via RT-PCR. Briefly, ovarian cortical tissues were flash-frozen and stored at −80 °C until RNA extraction using the RNeasy mini kit (Qiagen, Hilden, Germany), per the manufacturer’s instructions. Total extracted RNA was quantified using a NanoDrop™ One/One^C^ Microvolume UV-Vis Spectrophotometer (Thermo-Scientific, Carlsbad, CA, USA). Preparation of complementary DNA was performed using 200 (cat) or 100 (dog) ng mRNA with the Superscript II Single Strand cDNA synthesis kit (Invitrogen), following the manufacturer’s instructions. Reactions were performed according to the following parameters: preincubation at 95 °C for 10 min followed by 40 cycles of amplification with denaturation at 95 °C for 15 s; annealing at primer melting temperature (T_m_, 53–57 °C) for 30 s then extension at 72 °C for 30 s and finally melting at 95 °C for 10 s, 65 °C for 1 min, and 97 °C for 1 s. All reactions were performed in duplicate using a LightCycler^®^ 96 (Roche, Basel, Switzerland). Primer efficiency was assessed for each gene by serially diluting cDNA and used for analysis via the Pfaffl method [54].

### 2.7. Statistical Analyses

Data were analyzed in R Studio (version 1.3.1056) and presented as mean ± standard deviation. For Young’s modulus evaluations, tissue aspiration replicates were nested for each individual animal and set as a random variable, with treatment group (MMP exposure group) as fixed effect, and subjected to a linear mixed model in R Studio with the lme4 package [55] (v1.1-27) and with a post hoc Tukey test using multcomp [56] (v1.4-17). Collagen patterns were analyzed via a non-parametric Wilcoxon test using averaged results from image replicates for each individual, treatment, and day. Follicle density, stromal score, and gene expression data were analyzed via a non-parametric Wilcoxon test, with the significance threshold value set at *p* < 0.05.

## 3. Results

Tensile testing evaluations via micropipette aspiration (Figure 1a) of domestic cat ovarian tissue were performed. Prepubertal cat cortical tissues (<6 mo old, *n* = 11) averaged 16.9 ± 10.1 kPa (range 3.8–26.4 kPa), whereas adult individuals (6 mo to 2 yr old, *n* = 7) averaged 13.6 ± 10.1 kPa (range 4.9–31.7 kPa) (Figure 1b). For dogs, prepubertal females (<10 mo old, *n* = 16) displayed Young’s modulus means of 18.1 ± 10.0 kPa (range 7.4–32.9 kPa) and adults (10 mo to 3 yr old, *n* = 8) averaged 16.4 ± 4.7 kPa (range 9.2–26.2 kPa). Neither adult (with varying reproductive stages) nor prepubertal animals (with no evidence of ovarian antral follicles or corpora lutea, or uterine development in gross assessment) showed significant differences in Young’s modulus values between cats and dogs (Figure 1b). However, large inter- and extra-individual variations in Young’s modulus were observed among postpubertal animals of both species.

Next, cat and dog ovarian tissue sections were stained with picrosirius red, imaged under light microscopy, and analyzed for collagen fiber length, thickness, curvature, branch points, end points, alignment, lacunarity, and % high-density matrix (HDM) via TWOMBLI (ImageJ plugin, v.1). A comparison of fresh cat tissue patterns with fresh dog equivalents revealed significant differences in HDM (Figure 2), wherein dog HDM averaged 0.62 ± 0.14 (range 0.38–0.78) compared with cat HDM at about half that, at 0.33 ± 0.18 (range 0.14–0.63). Normalized branch points (number of branch points divided by total fiber length) and estimated fiber thickness (derived from HDM normalized by total fiber length) were also different between species, wherein cat ovarian ECM was characterized by an increased number of branch points and thinner fibers (*p* < 0.05, Figure 2b representative images). No differences in the other collagen metrics were observed.

Overall, the density of primordial follicles was higher in domestic cat tissues (13.7 ± 8.1/mm^2^) compared with dog tissues (7.8 ± 9.1/mm^2^), which was statistically significant in the absence of one outlier dog (5 mo old Boston Terrier, fresh tissue follicle density >2 standard deviations above the species mean, Figure 3a). No differences in transitional- or preantral-stage follicle densities were observed among species.

We then assessed the potential for matrix metalloproteinases and/or TIMP1 exposure to modify tissue tensile strength, collagen fibers, and subsequent follicle populations in in vitro culture. Following a 5-day incubation in either a control (unsupplemented), MMP1, MMP9, TIMP1, 1p1, or All treatments, cat and dog ovarian tissues were transitioned to the control medium and incubated until day 14. At days 5 and 14, tissues were again evaluated for elasticity/tensile strength (Young’s modulus), follicle density and collagen patterns, and gene expression.

In cats, ovarian Young’s modulus decreased in the MMP9 and 1p1 treatment groups compared with fresh controls (Figure 4a). This was maintained through the day 14 evaluation for MMP9. Average fiber lengths generally decreased in 14-day cultured tissues compared with fresh tissues (Figure 4b), which reached statistical significance for MMP9-, 1p1-, and All-treated tissues (*p* < 0.05). Only tissues supplemented with the MMP9 and All treatments also displayed significantly reduced fiber lengths between the day 5 and day 14 timepoints. Normalized branch points differed between fresh tissues compared with control, 1p1-, MMP9-, and All-treated tissues at 14 days of incubation (Figure 4b). Normalized end points were elevated between day 5 and day 14 of culture in all but the 1p1 treatment (*p* < 0.05). The fiber alignment score was elevated in day 14 cultured cat tissues compared with day 5 for 1p1. No statistically significant differences were observed in curvature, estimated fiber thickness, lacunarity, or HDM among treatment groups or timepoints (Appendix A). The densities of morphologically normal primordial- and transitional-stage follicles were maintained at 5 days of culture, but were significantly reduced by day 14 (Figure 4c). Preantral (primary and secondary)-stage follicle densities were negatively impacted by the MMP9, 1p1, and All treatments even in short-term incubation. Stromal scores were best maintained in day 5 control, TIMP1-, and All-treated cat tissues compared with fresh tissues (Figure 4d). Finally, while culture reduced the expression of all zona pellucida mRNAs, used as indicators of follicle density and development in cats, no statistically significant differences were observed among cultured treatment groups (*p* > 0.05, Figure 4e). Similar results were observed for the proliferation marker *PCNA* and oocyte quality marker *GDF9*, although no concomitant increases in *CASP3* were seen (*p* > 0.05). However, a slight elevation in cat *VEGF* expression was observed in all cultured groups compared with fresh controls (*p* = 0.042–0.058), but cultured tissues displayed no differences between each other.

As for dogs, no differences in tensile strength were seen among treatment groups compared with fresh controls at either time point, or between time points within the same treatment (*p* > 0.05, Figure 5a). Subsequently, no significant differences were found in any of the collagen metrics between fresh and treated tissues at either culture time (Figure 5b and Appendix A). Both normalized branch points and end points were increased in MMP9- and 1p1-treated tissues between days 5 and 14, and the fiber alignment score increased in the TIMP1 and All-treatments over the same period (Figure 4). A large variation in ovarian follicle density was observed among fresh domestic dog tissues, particularly with regard to a single outlier individual. While primordial follicle densities decreased with culture duration, no differences among treatment groups were observed. Transitional-stage follicle density was maintained in all treatment groups in short-term culture, with tissues in the 1p1 group maintaining density through day 14. Preantral follicles were similarly maintained in 5-day incubation in the presence of MMP1, TIMP1, and MMP9 compared with fresh tissue. Stromal cell scores were equivalent to fresh tissue in 5-day culture as well, and while some disorganization of the stromal cell compartment was observed in dog tissues at 14 days of incubation, they were not statistically significant (Figure 5d) compared with fresh tissues or among treatment groups. No statistically significant differences in *PCNA* or *CASP3* gene expression were observed in the dog tissues (Figure 5e), although similar patterns as in the cat culture of reduced *GDF9* (*p* < 0.05) and elevated *VEGF* (not statistically significant) were observed in cultured tissues compared with fresh controls.

## 4. Discussion

There has been increasing recognition of the impact of the ovarian physical environment on follicle and oocyte function in recent years. In the present study, we aimed to quantify the rigidity of domestic cat and dog ovarian cortical tissue and compare collagen patterns between the species. The overall goal was to improve our understanding of the ECM physiology underlying the differences in tissue stiffness and in vitro culture success observed among the species. Moreover, we evaluated the ability of matrix metalloproteinases and tissue inhibitors of matrix metalloproteinases to modulate cortical tissue rigidity, collagen content/patterns, and subsequent follicle density in vitro. This research had three main findings. First, dog tissues contain significantly more areas of high-density collagen matrix compared with cat tissues. Second, ovarian follicle density may have an impact on overall tissue tensile properties. Third, differences in MMP sensitivity were observed between dog and cat ovarian tissues, although exposure had a minor impact on follicle development in vitro.

Picrosirius red staining of cat and dog cortical tissues revealed a clear disparity between the species, wherein cat tissues displayed an overall lower-density collagen matrix. More detailed TWOMBLI analyses characterized the dog tissues as containing thick, well-aligned fibers with few gaps (lacunarity) and branching. Cat collagen fibers were characterized by thinner, more branched collagen compared with dog fibers. Work with in vitro-produced collagen matrices has demonstrated that fiber length and thickness, rather than simply total density, also modulates the mechanical properties of the collagen matrix [57]. Specifically, increased failure stress is correlated with increased fibril length and decreased diameter [57]. However, understanding how various collagen morphologies impact ovarian tissue tensile strength and cell behavior is still a growing area of research, particularly in the study of ovarian cancer [58,59], polycystic ovarian syndrome (PCOS) [60], and aging [28,61]—all of which are associated with changes in ovarian stiffness and/or fibrosis. For example, the ECM network of the human ovary was recently characterized via SEM, and it was found that fiber diameter and bundle thickness increase in reproductive-age females compared with prepubertal individuals but experience decreased porousness (most closely aligned with the lacunarity metric in the current study) in menopause [19].

Considering the significant differences in high-density matrices observed between dog and cat tissues, it was surprising that no corresponding differences in Young’s modulus values were seen between the two species. This may partly be attributed to the higher level of variability among dog tissues, which was one of the limitations of this study. While prepubertal dogs were identified based on ovarian and uterine morphology as well as age, the range of age at first heat is much more variable for dogs compared with cats. For example, small breeds typically experience their first heat by 6 months of age, but large-breed female dogs may not reach puberty until well after a year [62]. Moreover, the steps leading up to the first heat cycle are poorly defined in dogs [62], and as such, it is possible that some of the animals included in this study were nearing their first reproductive cycle. In women, a recent study evaluating the Young’s modulus of ovarian tissue via atomic force microscopy (AFM) demonstrated significant differences between reproductive ages, wherein prepubertal tissues were ~6.5 kPa, decreasing to ~3.2 kPa for reproductive-age ovaries, and increasing to 7.2 kPa for menopausal women [19]. Along these lines, reproductive cycling may explain the variability in post-pubertal dog tissues observed (Figure 1). Unlike cats, dogs experience long (up to 10 mo) periods of anestrus, wherein little to no follicle development and no ovulation occurs [63]—a state perhaps similar to prepuberty in terms of the ovarian microenvironment. Understanding how ovarian ECM and tissue viscoelasticity change between reproductive stages, potentially contributing to maintaining the quiescent state of the dog ovary during anestrus, would be valuable in decoding this phenomenon.

Interestingly, the recent AFM analyses of domestic cat ovarian tissues determined apparent elastic modulus values of 0.2 to 17.4 kPa [22]. While it is challenging to directly compare between techniques—even among AFM studies, due to differences in tip/probe material and size, etc.—and between differing modulus types, the Young’s modulus values obtained via micropipette aspiration in the current study overlap with, but are on average higher than, those reported for cats and other mammalian species. There are several reasons for this. First, the analysis region for AFM may range from 10 µm (as in the cat study [22]) to 100 µm (described in bovine work [21]), whereas the micropipette utilized here had an inner diameter of 500 µm. As such, the area evaluated in the current study represents contributions from a much larger cohort of ovarian components, which could either be seen as a limitation (less specificity) or an advantage of the analyses. The focal area in the present work also targeted a region of the ovarian cortex equivalent to ~250–750 µm below the ovarian surface. This likely encompasses areas of regional stiffness variation described by Hopkins et al. in the murine model, wherein the ovarian epithelium was found to be less stiff than the subepithelial zone (~200 µm below the murine ovary surface) and corresponding to an area with high follicular density [20]. Moreover, our technique also evaluates the tensile properties of tissues under stress, wherein they are actively stretched beyond the relaxed state via negative pressure.

It is also likely that there are species differences in cat versus dog ECM composition, as has been described in analyses of human, mouse, and sheep ovarian collagen. In the ECM analyses performed here, we did not evaluate additional ECM components, such as elastin and glycosaminoglycans, nor were we able to distinguish among collagen types. Previous work in the domestic cat has demonstrated that type I collagen is fairly uniformly distributed across the ovary, but that hyaluronic acid content is high in cortical tissues compared with the medulla, correlating well with that region’s increased rigidity [22]. Type I collagen has also been shown to be fairly well dispersed across follicular compartments in the murine [25] and ovine [64] ovaries, including being highly expressed in the murine ovarian surface epithelium [25], whereas immunostaining in the human ovary has shown to localize fiber bundles throughout the ovarian stroma, with little to no staining in the ovarian surface epithelium or granulosa cell compartments [27]. In mice, collagen type IV is primarily localized to the theca cell layer of secondary-stage and larger follicles, as well as the ovarian stroma. It has also been detected in the basal lamina in sheep [64] and humans [27]. Additional work is necessary to evaluate the ovarian localization of the major collagen types in cats and dogs, as well as the contributions of other ECM components supporting tissue elasticity.

It is interesting to note that a previous ultrastructural evaluation of domestic cat primordial follicles described them as containing basal lamina of “considerable thickness”, as compared with other species [65]. Oocytes themselves, in a bovine study employing micro-tactile sensors, were measured to have a Young’s modulus of 25.3 7.9 kPa [66]. Despite one outlier dog (with 27 follicles/mm^2^), primordial follicle density was significantly higher in cat tissues (avg. 13.6 follicles/mm^2^) compared with dogs (4.5 follicles/mm^2^, or 7.8 follicles/mm^2^ including the outlier). This inter-species difference has also been observed in previous work in our laboratory [67], although research in other labs have indicated that the overall preantral follicle population per ovary is higher in dogs [68,69] than cats [70], but these evaluations include additional follicle stages and consider the larger size of the typical dog ovary. As such, we postulate that the larger cat primordial follicle, and consequently oocyte, pool with substantive basal lamina may have also contributed to cat ovarian tensile strength in the absence of domestic dog levels of collagen HDM.

Additional research characterizing ovarian ECM patterns across species will be important for improving our understanding of the influence of soft tissue structure on its tensile properties and subsequent impact on follicular development. Mechanical loosening (via stretching to at least 10% beyond its original length) of human ovarian cortical tissue prior to culture has been proposed to simulate ECM remodeling/reorganization to promote follicle activation. Researchers found that the subsequently cultured tissues experienced a decrease in overall collagen content in a 6-day in vitro culture [71]. In the current study, we did not observe significant changes to bulk collagen following in vitro culture with MMPs in either species. These results are in contrast to a recent study in mice, where the treatment of ovaries from aged animals (with higher collagen matrix density and Young’s modulus compared with young ovaries) with collagenase (type IV, at 118.9U/mL for 1 h) decreased collagen density and restored tissue stiffness to levels similar to young mice. Although mouse ovarian tissue is also less rigid than that of larger species [72], short exposure to high collagenase concentrations may have been more advantageous in promoting bulk collagen reduction than the extended, lower exposures applied here.

Nevertheless, tensile testing results in our study demonstrated that domestic cat ovarian tissues were sensitive to gelatinase MMP9 exposure. This is supported by our observation that the reduction in fiber length with the corresponding increase in branch and end points in extended culture were consistent with the presence of MMP9 (MMP9 and All treatment groups). It was previously shown that 7 day treatment of cat ovarian tissues with 5 µM retinoic acid resulted in an elevation in *MMP9* and down-regulation of *MMP7* gene expression and increased proportions of activated (primary and secondary) follicles compared with fresh tissues and untreated controls [73]. Taken together, these data suggest an active remodeling of collagen in vitro in MMP9-exposed cat tissues. However, unlike in the retinoic acid supplementation study, we did not observe substantive changes in in vitro follicle activation via either gene expression or histological analyses for either cat or dog tissues. This was unexpected, particularly as type I collagen is generally abundant in mammalian ovaries, as described in [25,27,64]. One explanation is that, as MMP and TIMP expression is tightly controlled at both the transcriptional and protein (via inhibiting enzymes) levels [74], exogenous treatment may have been compensated for in these cultured tissues. Alternatively, one limitation of the current study was the use of human MMPs rather than feline- or canine-specific enzymes. While matrix metalloproteinase [75] and collagen [76] genes are highly conserved across vertebrates, and cat MMPs have had cross-reactivity with rabbit anti-human antibodies [35], the efficiency of human MMP activity on cat and dog collagens is not established.

We suspect that the increase in *VEGF* expression in both cat and dog tissues is an indicator that the agarose gel-based culture system was suboptimal for oxygen transfer in vitro, as *VEGF* is transcriptionally regulated by hypoxia-inducible transcription factor 1 (HIF-1) [77]. Hypoxia is associated with the development of fibrosis via HIF stimulation of the endothelial-to-mesenchymal transition and subsequent ECM deposition (kidney studies [78] and research on ovarian cancers [79]). Future work should explore the potential role of fibrosis in the relatively poor domestic dog ovarian tissue outcomes in vitro. Further explorations of methods to modulate the ECM to establish the optimal conditions for early follicle development in vitro are also warranted, either in native tissues as in the current study or in reconstructed systems, such as decellularized bovine ovaries used to improve our understanding of the various proteins contributing to cortical rigidity [21] and fibrin hydrogels recapitulating human ovarian ECM fiber patterns for isolated follicle incubation [80]. Cumulatively, this comparative biology approach with model mammalian species facilitates the understanding of the role(s) of various collagen types and structural arrangements in tissue stiffness and folliculogenesis, from the maintenance of the ovarian reserve to ovulation. This insight is important for establishing treatments to improve ovarian function in the human clinical setting, either via modulating the ECM in vivo or in the fabrication of biomimetic artificial ovaries to restore fertility or ameliorate the consequences stemming from the hormonal imbalances of menopause [81].

In sum, this study characterized, for the first time, the tensile properties of domestic dog ovarian cortical tissue. Although the differences in tissue stiffness among species were not substantiated in our analyses, the observed variances in ovarian collagen matrix density, fiber thickness, and branching provide future areas for investigation into the relative contributions of ovarian collagen characteristics to tissue rigidity and elasticity in these large mammalian models. Further, our results demonstrate that it is possible to slightly alter dog and cat ovarian tissue collagen patterns via MMP treatment in vitro, but that alternative exposure concentrations/durations may be needed to achieve tissue viscoelasticity modulation and subsequent effects on follicle development.

## Figures and Tables

**Figure 1 bioengineering-10-01285-f001:**
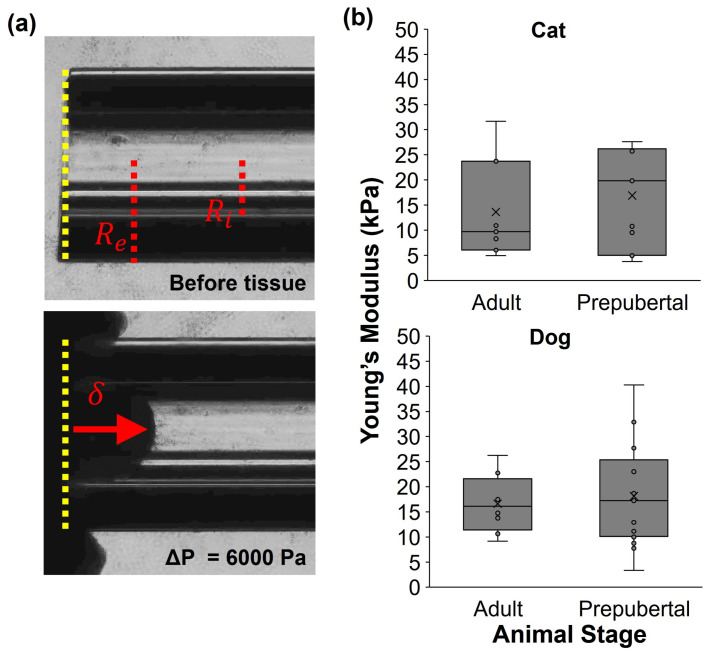
Characterization of dog and cat ovarian cortical tissue via micropipette aspiration, with (**a**) schematic of tissue aspiration technique, including yellow dashed lines indicating micropipette tip, red dashed lines indicating the inner and external pipette radii, red arrow demonstrating tissue distension under a given pressure (ΔP), represented by δ in the Young’s modulus equation, and (**b**) box-and-whisker plots of Young’s modulus values from prepubertal and adult domestic cat and dog fresh ovarian cortical tissue, where x represents the mean.

**Figure 2 bioengineering-10-01285-f002:**
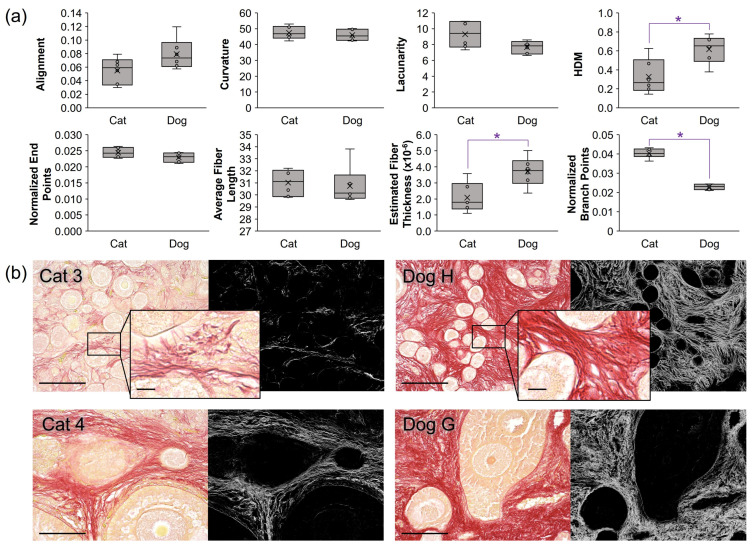
TWOMBLI characterization and comparison of domestic cat and dog ovarian tissue collagen patterns, with (**a**) box-and-whisker plots of fiber alignment scores, curvature, lacunarity, high-density matrix (HDM), average fiber length, estimated fiber thickness, and normalized branch points and end points where x indicates mean. Asterisks (*) denote differences between species with *p* < 0.05 following non-parametric Kruskal–Wallis test. (**b**) Representative images of picrosirius red-stained tissues from both species, with top row of images displaying tissues with high primordial follicle densities for each species, and lower row of tissues containing preantral and early antral-stage follicles; scale bars = 100 µm. Right panels represent HDM analyses for each corresponding PR image; inserts for cat 3 and dog H display 4x magnification (scale bars = 10 µm) of collagen fibers to display the less densely packed, thinner, and more branching fibers of cat tissue compared with dog tissue.

**Figure 3 bioengineering-10-01285-f003:**
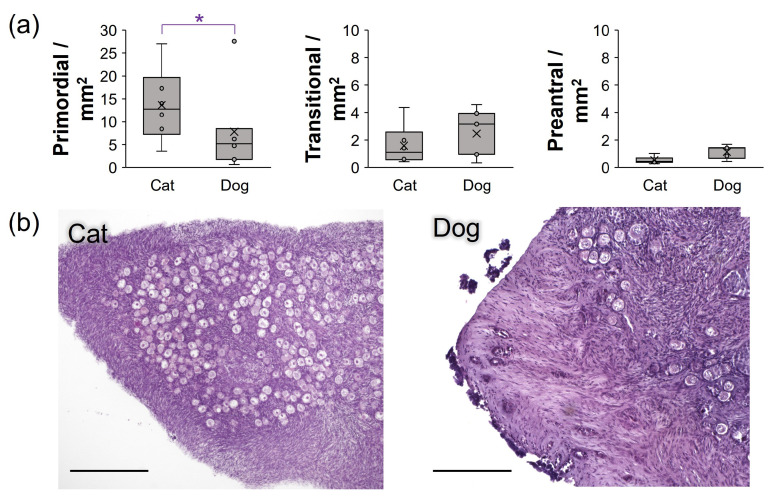
Density of follicles in fresh domestic cat (*n* = 6) versus domestic dog (m = 7) ovarian tissue in this study, with (**a**) box-and-whisker plots of primordial, transitional, and preantral -stage follicle densities where x indicates mean. Asterisks (*) denote differences between treatments, with *p* < 0.05 via non-parametric Wilcoxon test. Section (**b**) contains representative images of hematoxylin and eosin-stained ovarian tissues with higher densities of primordial follicles in cat sections, with scale bar representing 500 µm.

**Figure 4 bioengineering-10-01285-f004:**
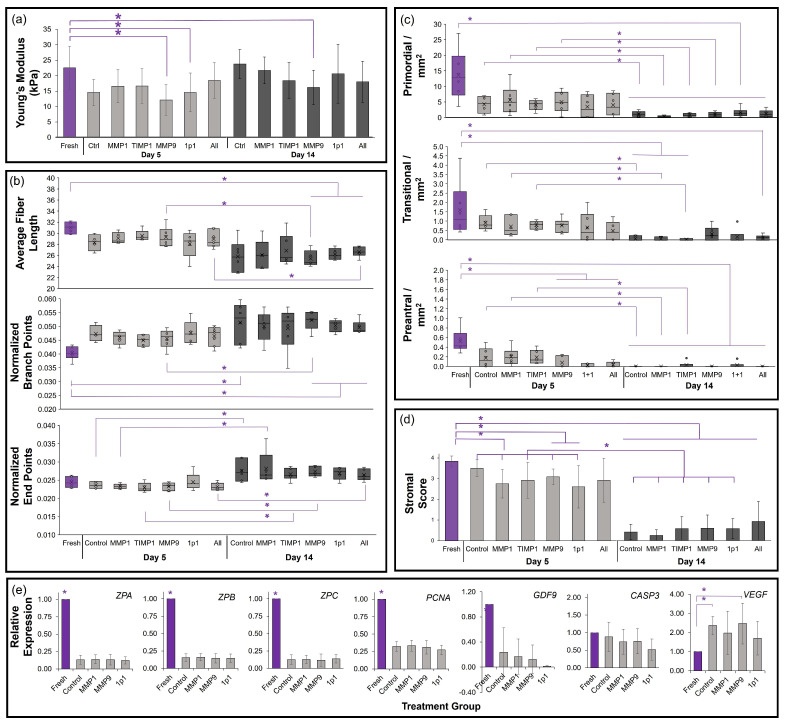
Influence of matrix metalloproteinase supplementation on domestic cat tissues (*n* = 6). (**a**) Bar graph of Young’s modulus via micropipette aspiration; (**b**) TWOMBLI collagen patterns box-and-whisker plots; (**c**) box-and-whisker plot of primordial-, transitional-, and preantral-stage follicle densities, where x indicate mean; (**d**) stromal score bar graph; and (**e**) gene expression bar graph. Treatments include fresh tissues (purple), compared with those after 5-day culture (light grey) under Control (no treatment), 30 ng/mL MMP1, 10 ng/mL TIMP1, 10 ng/mL MMP9, 30 ng/mL MMP1 + 10 ng/mL TIMP1 (1p1), or 30 ng/mL MMP1 + 10 ng/mL TIMP1 + 10 ng/mL MMP9 (All) treatments, followed by culture to day 14 (dark grey) in the control condition. Asterisks (*) denote differences between fresh tissues and treatments/days, or within treatments between days 5 and 14 of incubation, with *p* < 0.05 via non-parametric Wilcoxon test.

**Figure 5 bioengineering-10-01285-f005:**
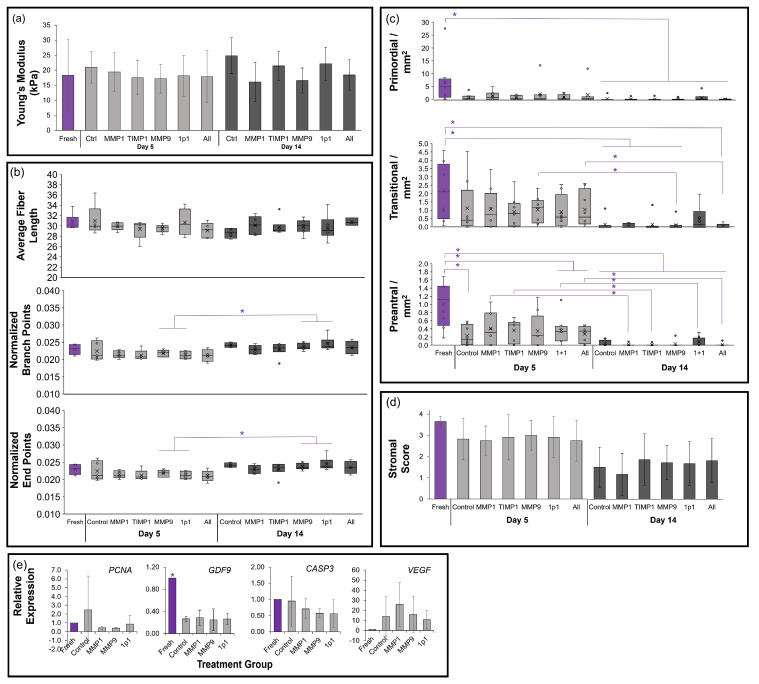
Influence of matrix metalloproteinase supplementation on domestic dog tissues (*n* = 7). (**a**) Bar graph of Young’s modulus via micropipette aspiration; (**b**) TWOMBLI collagen patterns box-and-whisker plots, where x indicates mean; (**c**) box-and-whisker plot of primordial-, transitional-, and preantral-stage follicle densities; (**d**) stromal score bar graph; and (**e**) gene expression bar graph. Treatments include fresh tissues (purple), compared with those after 5-day culture (light grey) under control (no treatment), 30 ng/mL MMP1, 10 ng/mL TIMP1, 10 ng/mL MMP9, 30 ng/mL MMP1 + 10 ng/mL TIMP1 (1p1), or 30 ng/mL MMP1 + 10 ng/mL TIMP1 + 10 ng/mL MMP9 (All) treatments, followed by culture to day 14 (dark grey) in the control condition. Asterisks (*) denote differences between fresh tissues and treatments/days, or within treatments between days 5 and 14 of incubation, with *p* < 0.05 via non-parametric Wilcoxon test.

**Table 1 bioengineering-10-01285-t001:** Primer details.

Species	Gene	Primer Sequence	Accession	Size	Efficiency
Cat	*ACTB*	F: ATCCACGAGACCACCTTC R: CACCGTGTTAGCGTAGAG	AB051104.1	75	2.00
	*CASP3*	F: CCACAGCCCCTGGTTACTAC R: TTCTGTTGCCACCTTTCGGT	NM_001009338.1	146	2.04
	*GDF9*	F: CATCCGTGGACCTGCTATTT R: CCAGGTTGCACACACATTTC	NM_001165900.1	129	1.96
	*PCNA*	F: TCTGCAAGTGGAGAACTAGGA R: GTTACTGTAGGAGAGAGCGGA	XM_003983740.5	170	1.99
	*VEGF*	F: CAGATGGAGAGCACAAACC R: ATACTCGATCTCATCAGGGT	XM_023253550.1	122	2.01
	*ZPA*	F: GGGAAGTTAAAATCTGTGAGC R: TGGTTTTGTCTGGTGACTG	NM_001009875.1	324	2.01
	*ZPB*	F: GGTGGCAGCTAGAGATGTGAG R: CGTTCTGTGGCGGATAGAGAC	XM_045039853.1	292	2.08
	*ZPC*	F: CAGGCCGAAGTCCACAC R: TCATGTTTCTGGGGTCATTAG	NM_001009330.2	238	2.33
Dog	*ACTB*	F: TCGCTGACAGGATGCAGAAG R: GTGGACAGTGAGGCCAGGAT	XM_845524.1	127	2.20
	*CASP3*	F: TTCATTATTCAGGCCTGCCGAGG R: TTCTGACAGGCCATGTCATCCTCA	NM_001419300.1	86	1.95
	*GDF9*	F: CAGAAGGGAGGTCTGTCTGC R: TGTTGGGGGAAAAGAAAGTG	NM_001168013.1	170	1.92
	*PCNA*	F: TCTGCAAGTGGAGAACTAGGA R: GTTACTGTAGGAGAGAGCGGA	XM_038571681.1	170	1.91
	*VEGF*	F: GTGCCCACTGAGGAGTTCAAC R: CCCTATGTGCTGGCCTTGAT	NM_001003175.2	72	2.20

## Data Availability

Data are available upon request.

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
