# Peer review of "Comparative Tensile Properties and Collagen Patterns in Domestic Cat (Felis catus) and Dog (Canis lupus familiaris) Ovarian Cortical Tissues"

_bioengineering, 2023, doi:10.3390/bioengineering10111285_

Round 1

Reviewer 1 Report

Comments and Suggestions for Authors

This is a well written article focusing on the difference of ovarian collagen fibers between cat and dog. The results can be used as a guide for the manipulation of ovary presevation. Only a minor question about the treatment by MMPs, due to the proteins were human orgin, the effects on cat and dog could be variable by the efficiency of proteins themselves, other than difference of collagen fibers between these two species.  

Comments on the Quality of English Language

No.

Author Response

To Whom it May Concern,

We would like to thank the reviewers for their time and constructive feedback on our manuscript. Please see below individual responses to comments, which have also been incorporated into the updated manuscript in blue font.

Sincerely,

Jen Nagashima, on behalf of all authors

________________________________________________________________________________________________________

REVIEWER #1

Comments and Suggestions for Authors

This is a well written article focusing on the difference of ovarian collagen fibers between cat and dog. The results can be used as a guide for the manipulation of ovary presevation. Only a minor question about the treatment by MMPs, due to the proteins were human orgin, the effects on cat and dog could be variable by the efficiency of proteins themselves, other than difference of collagen fibers between these two species.  

Thank you for your review. Regarding the use of human recombinant MMPs – MMPs genes are considered highly conserved across species, but there is variation in known protein sequences compared with human MMPs. That said, specific regions, such as MMP PEX domains (C-terminal haemopexin-like, involved in MMP localization / binding recognition) have indicated that some of these functionally relevant protein domains are conserved across several mammalian species (Cha et al 2002). Further, our laboratory group has previously utilized rabbit antibodies raised against human MMP1 to evaluate cat ovarian MMP1 localization (Fujihara et al 2016), which had positive cross-reactivity. However, the point regarding the efficiency of the proteins in degrading collagen fibers across species is well taken, and has been added to the discussion (Lines 520-4).

Reviewer 2 Report

Comments and Suggestions for Authors

This is an outstanding outcome from the authors. Well done. However, some minor corrections need to be made before can be accepted.

1. Ethical approval - even in vitro study, we need to provide an ethical study since you used animal tissues for the research. Please provide it.

2. 18 cats and 25 dogs - how do you justify this number in your research? Why not be consistent for both animals?

3. Tensile test - it is not clear how you conducted this test. There is no subtopic for this in the method section, please include it. And please state which standard (eg: ISO, ASTM etc) you refer to for the test, state the boundary conditions of the test, jig preparation and many more. This is really crucial part.

4. Figure 4a and 5a - i cannot see clearly the lower value of the average line. Please highlight it.

5. Figure 4e and 5e - too small. suggest enlarge it.

6. Figure 4b and 5b - the y-axis can be rescaled to the critical/desired value. Since the bar is too small, difficult to see it.

7. Limitations of the study should be included in the discussion section.

Comments on the Quality of English Language

Minor corrections 

Author Response

To Whom it May Concern,

We would like to thank the reviewers for their time and constructive feedback on our manuscript. Please see below individual responses to comments, which have also been incorporated into the updated manuscript in blue font.

Sincerely,

Jen Nagashima, on behalf of all authors

_____________________________________________________________________________________

REVIEWER #2

This is an outstanding outcome from the authors. Well done. However, some minor corrections need to be made before can be accepted.

  1. Ethical approval - even in vitro study, we need to provide an ethical study since you used animal tissues for the research. Please provide it.

Domestic cat and dog tissues were obtained opportunistically from shelter animals undergoing routine ovariohysterectomies at local spay/neuter clinics. These tissues were from un-owned animals, and would have otherwise been discarded. As such, no institutional review board permissions were required.  These details have been clarified in the text (Lines 109 and 563-66)

  1. 18 cats and 25 dogs - how do you justify this number in your research? Why not be consistent for both animals?

Thank you for this comment. The difference in sample size is primarily a factor of availability, as we obtained tissues opportunistically from local veterinary and spay clinics in their routine operations. While similar numbers ovaries of each species were available, we opted not to collect or analyze tissues from likely or known early-pregnant females, which was much more frequently observed in the cat population during the study period.

  1. Tensile test - it is not clear how you conducted this test. There is no subtopic for this in the method section, please include it. And please state which standard (eg: ISO, ASTM etc) you refer to for the test, state the boundary conditions of the test, jig preparation and many more. This is really crucial part.

Tensile testing details are under the subheading of “Micropipette aspiration”, which has been updated to “Micropipette aspiration tensile testing”. We have additionally included information on analyses (Lines 145-6, 153-7), and jig preparation (new Supplemental Figure 1, Line 129)

Biological tissues do not have standards defined by ISO, etc. because every tissue is different and NIST has not set any standard yet. We have used this method to measure the mechanical properties of cells and many others have used it to measure the mechanical properties of tissues and cells. Please see:

F.J. Byfield et al (2004) Biophys. J. 87:3336-3343

D.P. Theret et al (1998). J. Biomech. Eng. 110:190-199

Matsumoto et al (2002) Physiol. Meas. 23:635-648

  1. Figure 4a and 5a - i cannot see clearly the lower value of the average line. Please highlight it.

Figures 4a, c, and d as well as 5 a, c, and d are bar graphs, representing mean + Stdev, rather than box and whisker plots. We have added the lower Stdev line to these figures, and clarification of chart type has been added to the figure legends

  1. Figure 4e and 5e - too small. suggest enlarge it.

These subsections have been enlarged

  1. Figure 4b and 5b - the y-axis can be rescaled to the critical/desired value. Since the bar is too small, difficult to see it.

The y-axes were scaled to be better comparable between cat and dog results for each criteria. Some of these (specifically the normalized branch and endpoints) have been modified (not starting at 0) to increase visibility.

  1. Limitations of the study should be included in the discussion section.

The limitations of the study are scattered throughout the discussion section; however, we have adjusted the text to explicitly / more clearly note these and expand the discussion further (see Lines 424, 451-2, 461-3, and 520-4).

Reviewer 3 Report

Comments and Suggestions for Authors

Nagashima et al conducted studies comparing some features of ovarian cortical tissues between cat and dog. They demonstrated that dog tissues contain significantly more areas of high density collagen matrix compared with the cat. And, they suggested that ovarian follicle density may have an impact on overall tissue tensile properties. Furthermore, differences in MMP sensitivity were observed between dog and cat.

Overall, the manuscript is well-written, and results of experiments are interesting. I have only one comment. Please add more descriptions about why comparing features of ovarian cortical tissues between cat and dog is important, especially focusing on thinking about human clinical settings.  

Author Response

To Whom it May Concern,

We would like to thank the reviewers for their time and constructive feedback on our manuscript. Please see below individual responses to comments, which have also been incorporated into the updated manuscript in blue font.

Sincerely,

Jen Nagashima, on behalf of all authors

________________________________________________________________________________________________________

REVIEWER #3

Comments and Suggestions for Authors

Nagashima et al conducted studies comparing some features of ovarian cortical tissues between cat and dog. They demonstrated that dog tissues contain significantly more areas of high density collagen matrix compared with the cat. And, they suggested that ovarian follicle density may have an impact on overall tissue tensile properties. Furthermore, differences in MMP sensitivity were observed between dog and cat.

Overall, the manuscript is well-written, and results of experiments are interesting. I have only one comment. Please add more descriptions about why comparing features of ovarian cortical tissues between cat and dog is important, especially focusing on thinking about human clinical settings.  

Thank you for your review. An additional note on the application of the knowledge gained by these species comparative analyses for human clinics has been included in the manuscript, Lines 537-43. Specifically, this comparative approach can provide insight into how different ECM molecule combination and structures influence tissue stiffness. In turn, this information has value in developing methods to modify human ovarian tissue rigidity and/or develop in vitro ECM gels or even artificial ovaries which recapitulate folliculogenesis-optimizing ECM structures for fertility preservation efforts. These techniques are of particular interest for patients with ovarian dysfunction (cancer, PCOS), or those in need of fertility preservation options (undergoing fertility-affecting treatments, or aging women).